# Greek Pig Farmers’ Perceptions and Experiences of Tail Biting and Tail Docking

**DOI:** 10.3390/ani13040672

**Published:** 2023-02-15

**Authors:** Michail Kakanis, Katerina Marinou, Evangelia N. Sossidou

**Affiliations:** 1Department of Veterinary Service, Regional Unit of Pieria, Directorate of Rural Economy and Veterinary Service, 25 Martiou 49, 60132 Katerini, Greece; 2Directorate of Animal Welfare, Veterinary Medicines and Veterinary Applications, Hellenic Ministry of Rural Development and Food, 2 Acharnon Street, 10176 Athens, Greece; 3Veterinary Research Institute, Ellinikos Georgikos Organismos-DIMITRA, 57001 Thessaloniki, Greece

**Keywords:** tail biting, pig farmer attitudes, pig welfare, tail docking, risk factors

## Abstract

**Simple Summary:**

Tail biting is an abnormal behavior of pigs, posing serious welfare and economic challenges in intensive pig farming. There are no studies on the perceptions of farmers in south European countries on tail biting and tail docking. This study aims to examine the attitudes of Greek pig farmers toward tail biting and tail docking through a structured questionnaire. In case of an outbreak, the Greek farmers mainly choose to remove the victim, while 64.4% have tried at least once to raise pigs with intact tails. Chains are still the most commonly used enrichment material. Feed quality, environmental factors, and health problems are considered the most important risk factors. The study indicates that solutions for effectively reducing the need for tail docking should consider farmers’ perceptions and everyday practices.

**Abstract:**

Tail biting is both an important economic and a welfare problem. The primary preventive measure, almost on a routine basis throughout Europe, remains tail docking following a risk analysis. This study aimed to get insight on the perceptions of Greek pig farmers towards tail docking, intervention measures, and risk factors of tail biting, as well as opinions on pig welfare themes. Eighty-two farmers answered a questionnaire provided online and in printed form. In the case of a tail-biting outbreak, the most important intervention measure is the removal of the bitten pig (victim), while feed quality, air movements, and stocking density were ranked as the highest risk factors (*p* ≤ 0.05). Chains are the most common type of enrichment by 67% followed by plastic objects by 29.5%. Almost half of the farmers reported having no tail-biting problem on their farm, while 64.4% of the respondents have already tried at least once to raise pigs with intact tails. To reduce routine tail docking, it is essential to apply specific farm-oriented solutions effectively. To succeed in this aim, it is important to consider farmers’ concerns and practices. This study is the first step in that direction.

## 1. Introduction

Tail biting poses a significant risk to the welfare of pigs in modern intensive farming, resulting in serious economic losses [1] while threatening the economic sustainability of the pig farming sector [2]. This abnormal behavior can cause acute pain in bitten pigs [3]. Still, it may also lead to long-term consequences such as secondary infections [4], carcass abscesses [5,6], respiratory organ inflammation [7,8], and pulmonary pathologies [9]. Furthermore, tail biting indicates reduced welfare also for the pigs performing this behavior. Studies have suggested that environmental inadequacies can lead to frustration and discomfort in pigs. It has been supported that tail biters become tail biters possibly due to experiencing chronic stress [10]. Due to the multifactorial basis of this abnormal behavior, scientific research has proven a plethora of risk factors; however, it is challenging to identify the single one that usually triggers an outbreak. This results in more than 95% of pig farmers throughout Europe tail docking, most on a routine basis [11], although Directive (EC) 2008/120 [12] prohibits this practice. In practice, tail docking mitigates the risk but does not eliminate it [13], as can be seen by a high tail-lesion prevalence, as much as 72.5%, in tail-docked populations in abattoirs [1]. Apart from the ethical issues raised [14], tail docking provokes acute and long-term pain [15] and, used on a routine basis, could possibly hide more serious chronic welfare problems on the farm [16].

A vast number of factors have been proven to play a role in triggering an outbreak, such as poor health [17], stress [8], feed [18], barn climate [4], breed [19], stocking density [20], and lack of appropriate enrichment material [4]. Thus, there is no single solution that can be applied to a pig farm, making farmers hesitate to implement the necessary changes [21]. Although the European Food Safety Authority (EFSA) has used 25 different hazards in risk assessment models [22], the preventive measures must be on a single-farm basis, chosen by efficacy and the initial prevalence of tail biting [2]. It is necessary to examine the relation between farmers’ acceptance of tail biting and their previous experiences dealing with this problem [23]. On the other hand, farmers have different views on tail-biting problems than scientists due to their everyday practical management of their pigs [24]. Furthermore, farmers’ perceptions from different countries can differ not only on practical grounds but also due to traditional or societal reasons [25]. Finnish producers rank higher the use of bedding material as an interventive [16] or preventive [26] measure while Irish producers rank it last [27]. Until now, very few studies explore farmers’ perceptions [25] but to best of our knowledge all have been performed in Northern European countries which, to a greater or lesser extent, refer to the same infrastructure and production systems (Appendix A).

As tail biting is high on the welfare agenda of the European Commission, there is a growing pressure on Member States to fully enforce Directive (EC) 2008/120 and implement a national action plan towards raising pigs with intact tails. The success of any such plan should take into consideration farmers’ perceptions [25].

The aim of this study is to give an insight into Greek pig farmers’ opinion on tail biting, tail docking, and practical management in the case of an outbreak. Furthermore, the aim is to obtain knowledge about the risk factors they consider important, the kind of enrichment material they use, as well as their attitudes towards pig welfare. In Greece, most farms are small- to medium-sized (avg. 200 sows), while the vast majority are farrow-to-finish farms, raising pigs indoors. Ad libitum dry feeding is used rather than liquid diets, while floors are partly or fully slatted, and most farms use a combination of mechanical and natural ventilation. This is the first report on pig farmers’ views coming from a South European country, where not only is the infrastructure different but the climatic conditions and especially challenges in the context of ongoing climate change are also different.

## 2. Materials and Methods

Commercial pig farmers were invited by the local veterinary services to complete a printed version of a structured questionnaire survey. The survey was created in collaboration with veterinarians specialized in pig internal medicine and researchers from the veterinary research institute of Greece named ELGO-DIMITRA, drawing inspiration from a previous study by Valros et al. [16]. A preliminary version of the survey was shared with a small group of pig producers, and any questions found to be unclear were modified. Furthermore, an online version of the questionnaire based on Google Forms was uploaded and further promoted with an explanatory note on the scopes of the study on the site of the Greek National Pig Association [28]. Moreover, the printed survey helped us not to have a biased sample of technology-friendly farmers as it allowed those unfamiliar with internet technologies to express their views too. The data collection took place from September 2018 to May 2019 while all paper questionnaires’ data were entered into Google Forms in order to have all answers in one single data frame. All farmers were informed and gave their consent on the use of their personal data under provisions of regulation (EU) 2016/679 [29].

The questionnaire (Appendix B) comprised of three main sections. The first section included questions about the farm identity and production parameters regarding herd size, number of piglets per sow, etc. Furthermore, farmers provided data about the type of ventilation and flooring, feed type, number of animals per pen, and number of drinkers per pen.

In the second section of the questionnaire, farmers were asked to assess the status of tail-docking and tail-biting prevalence on their farm. Specifically, they were asked about the length of the remainder of the docked tail on a scale from ¼ to intact. Respondents were also asked what percentage of tail biting they considered to be a significant welfare problem on their business. Additionally, this section included questions about the type of enrichment farmers use and how useful they consider some common managerial practices in case of a tail-biting outbreak (from 0 = “not useful at all” to 5 = “very useful”).

On the third section of the questionnaire, farmers were asked to rank the importance of some risk factors for a tail-biting outbreak. In addition, there were questions about their level of agreement/disagreement (from 0 = “strongly disagree” to 5 = “strongly agree”) on statements about concepts of pig welfare. In case of intervention measures and risk factors, the above scale was transformed in a 5-point Likert type scale, combining 0 and 1 in a new variable to have a more concrete result. In order to report frequencies in some cases, 1 and 2 are considered “Disagree”, 3 considered “Neutral”, and 4 and 5 considered “Agree”. Producers were also given the opportunity to write open comments about some of the aforementioned topics.

Data analysis was performed using IBM SPSS Statistics for Windows, version 27 (IBM Corp., Armonk, NY, USA). As most variables were non-normally distributed and many were non-continuous, statistical tests were performed using non-parametric tests. Descriptive statistics were used to score the importance of intervention measures and risk factors for tail biting, given as mean and standard deviation. The ranking was based on mean values to discriminate differences in data better than the median, which does not discriminate differences well in data based on a categorical scale. Friedman’s two-way analyses, followed by the Wilcoxon signed rank-sum test for pairwise comparisons, were used to test differences in the mean score given to preventive measures or risk factors when appropriate. Spearman rank correlations were used to analyze farm size, perceived seriousness of biting, and acceptable levels of biting. The same was utilized to analyze the relationship between tail length, the perceived severity of biting, and the level of biting that is considered acceptable.

Significance level was accepted at *p* = 0.05.

## 3. Results

A total of 82 pig farmers from 18 regions around the country participated in the survey.

In Greece, pig farming is mainly placed in four regions that account for over 50% of the total number of sows, and they are well-represented in this study. All farms that took part were of the type ‘farrow to finish’. The size of the farms was stratified, as shown in Table 1, and varied between 17 sows to 1980 sows, while the average herd size (234 ± 276 sows) was representative of the national average. Greece is considered to have approximately a population of 55,000 sows (data from the National Pig Association), so roughly 35% (19,201) of the sows farmed were represented in this study.

The online survey was chosen from half of the responders. This shows that, slowly but steadily, pig farmers embrace the use of new technologies for their information, which is a point that any future policymakers should consider.

Of the farmers, 25.3% run their businesses on a family basis without an employee, while 60% of farms were built before 1990; 72.3% of workers had more than seven years of experience with the same employer. This suggests that workers probably have a good level of knowledge of tail biting. On the other hand, workers for pig farms are of limited availability in the pig industry in Greece. This is further supported by the fact that over 74.2% believe the scarcity of highly skilled workers is a problem for pig welfare in the industry. The average pen size in weaning is about 20.9, while in the late grower phase, it decreases to 18.2, ranging from 6 up to 70. The vast majority of farmers (81.0%) use dry feed, while pellets are fed only by 5.1%. Fully or partly slatted floors were used by over 92% of the respondents in all three stages of production (weaners, growers, and finishers). Most farmers perform the teeth-clipping, castration, and tail-docking procedures in the first three days of a pig’s life. Interestingly, only one respondent out of 81 uses immunocastration, while all others use surgical castration. More than half of the respondents (55.6%) leave 1/3 or less of the tail (Table 2). Tail docking is a widespread practice as in the rest of Europe; however, 9.9% of farmers who took part in the survey do not perform tail docking.

Seeing “blood on the tail’’ is the most important sign for a producer to intervene, probably because bite marks on the tail are difficult to distinguish from outside the pen during the daily assessment of pigs (Appendix A). Chains are the most frequently used enrichment material (67.1%), followed by plastic objects (29.5%). Here is another discrepancy with the EU regulation, where chains are identified as a suboptimal enrichment material that has to be accompanied by an optimal one. The onset of tail biting for 54.7% of the respondents was reported to occur in the grower phase. While 64.4% of farmers have already tried, at least once, to raise pigs with intact tails, unfortunately, they encountered serious problems with tail biting, as high as 60%.

Of the farmers, 76.9% responded that they have no tail-biting problem, while 8.98% consider tail biting a very important problem on their farm (Table 3); 52.11% consider that having 1–2% of tail biting would pose a serious problem for their farm (Table 4).

There was a statistically significant correlation between the level of severity of the tail biting on the respondents’ farms and the percentage of tail biting considered as serious for farm management; thus, the more tail biting observed on a farm, the higher level of tail biting was considered manageable (r = 0.386, *p* = 0.001). The length of the tail after tail docking was negatively correlated with farm size, meaning that large farms left a smaller part of the tail after tail docking (r = −0.387, *p* < 0.001). There was no correlation between farm size and tail-biting prevalence on a farm or the percentage of tail biting in a farm accredited as serious.

The study participants were asked to rate different actions they would take during a tail-biting outbreak on a scale of 1 to 5, with 1 being “Doesn’t help at all” and 5 being “It helps a lot.” Removing the bitten pig was considered the most critical intervention measure in case of a tail-biting outbreak. It was ranked higher than all other measures (*p* < 0.05 for all comparisons). Removing the biter was the second intervention measure to follow, possibly due to difficulties identifying the biter. It was ranked higher than all other measures (*p* < 0.05 for all comparisons). Enrichment materials were not considered to be very helpful while adding pig lick blocks received the lowest mean score. Adjusting the ventilation, adjusting the temperature, and reducing the stocking density were ranked significantly higher than reducing light or putting a pig lick block (*p* < 0.05 for all comparisons). There was no difference in the scoring of intervention measures between farms with different docking lengths (Appendix A).

Not all of the risk factors included in the questionnaire were perceived as being particularly important by the respondents, as ten of them scored lower than 2.5 on average (Table 5). Feed quality was the top-rated measure. Stocking density and ventilation, which are factors linked to farm management, came in close behind as important risk factors. Farms with different docking lengths had no difference in their rating of risk factors.

The only risk factor that differed significantly in correlation with farm size was pen hygiene (Kruskal–Wallis test *p* = 0.019) (Figure 1) and a tendency for health status (Kruskal–Wallis test *p* = 0.061).

Of the respondents, 68.4% believe there are no other ways than tail docking to prevent tail biting, while more than half of them could use economic incentives to raise pigs with intact tails (Table 6). There was a high level of consensus, 91%, between farmers, that in the Greek meat market, there is no willingness from consumers to pay for meat from pigs with intact tails.

## 4. Discussion

The aim of the present study is to provide for first time a detailed picture of Greek pig farmers’ attitudes towards tail biting and tail docking. The rate of responses submitted online was unexpectedly high, compared to other countries’ experience [30].

In the current study, the first action to take during a tail-biting outbreak for most respondents is the removal of the bitten pig, followed by the removal of the biter. This is probably related to the significant time needed for a farmer to examine the pen and identify the biter carefully. The results follow the same ranking as in the study of Hunter et al. in 2001 [18] and the Irish study [27]. In the Finnish [16] and the Dutch study [24], on the contrary, removing the biter was the first choice of action. There are not many studies that explore how to solve a tail-biting outbreak effectively. Zonderland et al. [31] found no difference between removing the biter and adding additional straw. Straw addition at the start of an outbreak mitigates the risk of an escalation but cannot effectively stop the behavior [32]. Furthermore, Chou et al. [33] found that the ratio of biters/victims in the pen is more critical for dealing successfully with the outbreak than the intervention methods selected. Removing the victim increases the need for more hospital pens, while removing the biters has the potential to start a new outbreak. The use of enrichment material was not ranked as high as in the Finnish study [16], probably because in Greek farms, caretakers have fewer choices due to the fully slatted floors and their concern with blocking the slurry system. This means that scientific research must consider the possible limitations in the available materials and give concrete solutions that farmers are willing to adopt and able to follow in their everyday practice. Reducing animal density and taking care of a stable microclimate comes next as an intervention measure. At the same time, the use of anti-biting substances and pig lick blocks was ranked very low in this study, in agreement with the answers from studies in other European countries. This could be attributed to the increased workload required for using anti-biting substances while the use of pig lick blocks could possibly take a long time to correct any possible feed deficiency.

The respondents in this study ranked the bad quality of feed higher than any other risk factor. This is in accordance with the results from other studies where feed quantity and quality problems are ranked in the first ten preventive measures, except the Dutch study where the feeding system and feed were last in a list of ten risk factors. Imbalances in the diet could increase foraging as pigs try to get the correct balance of nutrients [34], driving them to tail biting. The over- or under-supply of nutrients could result in an imbalance in essential amino acids, gut microbiota, and/or immune response in specific pigs in the pen, thus risking the development of tail biting [35]. Furthermore, mycotoxins in the feed can activate the immune system, affect neurotransmitter systems, negatively influencing mood, and provoke anorexia [36]. Greek producers ranked ‘Feeding always at the same time of the day” as the 15th out of 20 risk factors, considered less important, as did the UK (17th) and the Irish producers (19th), but not the Finnish producers (10th). The different feeding systems used in the two countries could explain this. In Greece, ad libitum feeding is predominant, while Finnish farmers have long troughs and know how important it is to reduce competition at feeding.

Farmers ranked stocking density as a risk factor in second place. It is noteworthy that the same rank was also found in the UK, Irish, and Dutch studies, with only the Finnish study ranking it in the 11th place. This probably reflects to a large extent the fact that stocking density in Finland is even lower than required by the EU legislation. High stocking density as a tail-biting risk factor interferes with normal social interactions [17], but it is not well-documented in experimental studies [37]. Moreover, it has to be considered that group size and space allowance are confounded and probably interact with other factors [38]. Although stocking density is considered by pig farmers to be an important risk factor for tail biting, it is not ranked as high as a prevention measure. This could be explained by the fact that, in case of an outbreak, there probably is not enough empty cells to move all pigs without initiating a new outbreak. As the birth rate and viability of newborn piglets significantly increase in the intensive pig industry in rather old buildings, producers will have to consider constructing new buildings or reducing the number of sows to overcome density problems.

Any inability to keep a stable microclimate in the barn was ranked among the ten most important risk factors (Ventilation 3, Temperature 5) in the current study. This applies to all countries that participated in similar studies even though they have different farming systems, and is in contrast to the lower ranking of this risk factor by EFSA [22]. Keeping pigs out of their comfort zone creates discomfort and chronic stress to the pigs, leading to tail-biting outbreaks [4]. Moreover, as climate change advances, this factor will probably receive more attention, as there are limits to the capacity of ventilation/heating/cooling systems [37].

Although animal health problems have always been a priority for Greek farmers as a risk factor for tail biting, there are not enough research studies on the topic. The link between poor health and tail biting is yet to be discovered, but there are indications of a two-way causal mechanism [39]. Poor health provokes tail biting, possibly through the production of pro-inflammatory cytokines, that induces sickness behavior and influences the balance of neurotransmitters regulating the stress response. Through tail biting lesions, pathogens enter the body and can cause health problems evident in carcasses as lung lesions, abscesses [5], and osteomyelitis [6]. Moreover, as health problems and tail-biting lesions require antibiotics, it is essential to take early care of any health issue in the context of reducing the use of antibiotics. The latter was one of the reasons justifying the use of tail docking by Finnish farmers if it was legal [16].

The lack of appropriate enrichment material as a risk factor for tail-biting outbreaks is not considered so important and there seems to be a consensus among farmers in the European region. Although a vast number of scientific research has been focused on using enrichment material, considered the cornerstone of tail-biting prevention and intervention by experts [23], farmers do not adopt these views. This is supported in this study, where almost 60% of Greek farmers disagree that the boredom of animals is a welfare problem in the Greek pig industry, a result that is in accordance with the Dutch study [24]. EFSA scientific opinion ranks access to manipulable objects as the most important of all [23]. This discrepancy could be attributed up to a point to certain limitations on the use of bedding imposed by using slatted floors in piggeries, but it remains a point of concern. It demonstrates that there is an information gap between scientists and farmers that must be bridged. As farmers have a different context of animal welfare, based more on an economic basis [40], it is crucial to understand their point of view on the practical use of enrichment material. A thorough understanding of the needs of animals and the correct method of care is essential for those who work with them. With appropriate training, considering farmers’ perceptions and holistic views due to their everyday experience can help ensure that animals are treated respectfully, which is essential for ethical and legal reasons.

Tail length was ranked as the fourth risk factor in this study and fifth in the Dutch one [24]. Studies confirm this as long-tailed pigs have more tail lesions than short tails docked in farm experiments [41]. Tail docking remains a reasonably effective solution at a low cost [14]. Although light intensity is more immense in Greece, it was not ranked high as a risk factor, nor are breed and sex. In the study of Valros and Barber [25], breed was ranked relatively high at position ten, probably since UK farmers have a lot of local breeds that they raise, while in Greece, they are raising mainly crossbreds.

According to the results from studies in countries that tail-dock, an average of 76.5% of the respondents consider tail docking a necessity in the modern pig industry. This could be based on their current knowledge and day-to-day management as they are not used in raising pigs with intact tails, and farmers’ views are influenced by the system they work with and they tend to defend the production system they use [42]. Moreover, their belief could be further amplified by the relatively high percentage of respondents that had already tried to raise pigs with intact tails—more than sixty percent, as opposed to 34% in the Dutch study [24] and four farmers in the UK [25]. Unfortunately, on all occasions, there were serious problems with tail biting, probably cementing farmers’ belief that tail docking is the only way to avoid tail biting. The high percentage of respondents trying at least once to raise pigs with intact tails suggests that farmers want “a change” in their industry and are willing to take on initiatives provided that are well-organized and with the appropriate education. Solutions that consider farmers’ perceptions about practicability and effectiveness would possibly be adopted more easily [43]. Their will to change is further supported by the answer to the question “Do you believe it is better to dock all tails than to run the risk of tail biting, even if it concerns just one bitten pig?”. 58.2% answered that they would not do that, indicating that farmers want to comply with legislation and accept that a certain percentage of tail biting will always be present sporadically on the farm. In the same context, the number of people that had already tried to raise intact pigs could explain why most respondents keep little of the tail, as, due to problems they encountered, they are keener on keeping the tail as short as possible [25].

Additionally, farm size seems to be linked to the size of the tail docking. The bigger the farm, the bigger portion of the tail is docked. This link also appears in the study of Dutch farmers [24]. This practice could be applied by more prominent farmers to confront the inevitabilities of dealing with a problem in vast numbers. This is further supported by the fact that both studies ranked tail length as a risk factor relatively high.

Most respondents report that they do not have a serious problem on their farm, but in a pilot study undertaken in Greece in the framework of the COST Action CA15134 “GroupHouseNet,” the prevalence of tail biting lesions in 2017 and 2018 were 46.4% and 51.0%, respectively, despite tail docking [44]. Interestingly, there was a positive correlation between the tail-biting prevalence on the farm and the percentage of what was considered “a serious problem”. The bigger the problem in a farm, the bigger percentage is accepted as a problem.

Health problems seem to be a significant welfare concern for Greek farmers, probably due to farmers’ and scientists’ different perception of welfare. Farmers have an economic context when they refer to animal welfare [40] and consider it mainly good health [42]. Additionally, dealing with an active outbreak is complex, and success is not coming quickly, while failure reduces the satisfaction from work for the farmer and his faith that he can deal with the problem. Farmers believe that scientific facts remain too unclear for implementation and only serve as guidance [40], while this was further supported by the Irish study [27], where all farmers during the interview reported that no clear-cut solution exists. Although scientific research suggests that pain is not only short-lasting but probably even long-lasting [15], producers feel that the pain associated with docking is minimal [42], a fact that our study seems to support as more than 2/3 (70.00%) of the participants disagree with the statement “Tail docking is painful”.

As most Greek farms were constructed back in the 1980s, every probable management decision comes at a higher cost as necessary renovations should be considered. Production changes may be needed to implement a ban on tail docking [6] successfully and this could explain why 53.2% of farmers believe that they should receive economic incentives.

The multifactorial origins of tail biting are further supported by the variety of answers in the open questions’ comments. Interestingly, some respondents report that tail biting is “in the nature of the specific pig”. Research has suggested that individual influences and specific neurobiological characteristics play an essential role in some individuals becoming tail-biters [45].

In this study, some limitations were identified. No demographic data or information on the degree of training of the farmers is currently available to the research team, which would add an interesting insight. Additionally, responses were received from only some local regions, resulting in some areas of the country being underrepresented. However, the production regions that account for more than 50% of Greek production were represented. Furthermore, farms specializing in fattening or piglet production did not participate in the study. Still, it should be noted that fattening or piglet-producing farms are not typical in the Greek farming system.

The study also highlighted the need for farmers’ training to be tailored to their experiences, the farming system employed, and the management practices followed. Furthermore, economic incentives and networking could aid farmers in reducing the risk of tail biting and the widespread practice of tail docking.

## 5. Conclusions

Greek pig farmers’ perceptions and experiences of tail biting and tail docking have been investigated following previous studies showing the high prevalence of this behavior in Europe. Specificaly, Greek farmers appear to underestimate the importance of tail-biting than producers from, Finland. Nevertheless, Greek farmers ranked measures in the same order as producers from other countries. They identified feed type and stable microclimate as the most critical risk factors for an outbreak of tail biting. Removal of the victim pig was the most common intervention. Understanding the concerns and practices of the Greek producers should be further exploited to organize awareness campaigns and training programs to support the Greek pig industry in complying with EU legislation and help improve the animals’ welfare status.

## Figures and Tables

**Figure 1 animals-13-00672-f001:**
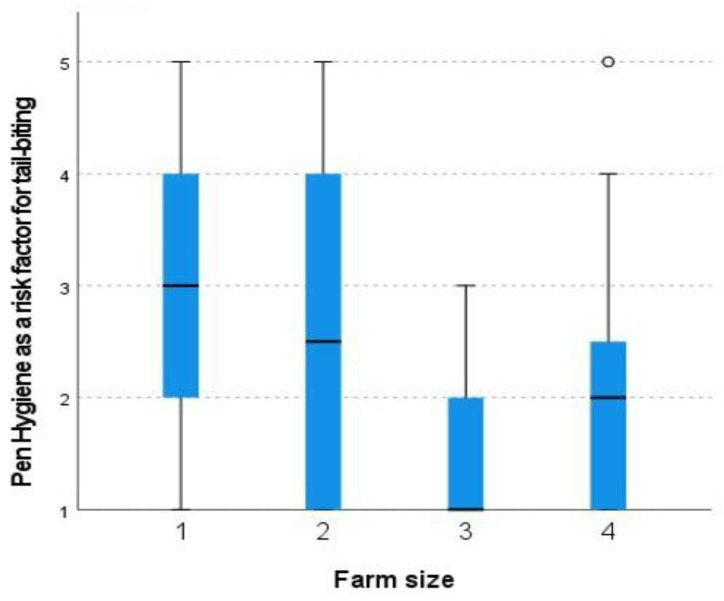
Scoring of the importance of pen hygiene as a risk factor for tail biting, on farms with different size. (Number of sows: 1 = 17–99, 2 = 100–299, 3 = 300–499, 4 = 500+). Scale (0 = Not important at all, to 5 = Very important). The boxplot indicates medians and quartiles (25% and 50%) as well as outliers.

**Table 1 animals-13-00672-t001:** Distribution of farm size according to number of sows. In parenthesis, the number of respondents from farms located at the main pig-producing regions.

Farm Size	Number of Farms	Percentage
17–99	29 (13)	35.3
100–299	31 (20)	37.8
300–499	11 (7)	13.4
500–1980	11 (7)	13.4

**Table 2 animals-13-00672-t002:** Number and percentage of farms according to tail length after tail docking.

Tail Remnant	Number of Farms	Percentage
1/4	20	24.7
1/3	25	30.9
1/2	22	27.2
3/4	6	7.4
No docking	8	9.9

**Table 3 animals-13-00672-t003:** The level of severity of tail biting (TB) on farm. Scale: 0 (not a problem at all) to 5 (very important problem).

Scale	TB Problem on Farm
0	48.71% (38) *
1	28.20% (22)
2	6.42% (5)
3	7.69% (6)
4	5.13% (4)
5	3.85% (3)

* Data are given as a percentage, and in parenthesis, the number of replies.

**Table 4 animals-13-00672-t004:** Presentation of level of tail biting (TB) that farmers consider to be a “serious problem” for their farm management practices.

Percentage of TB	What Percentage Do You Consider as “Serious TB Problem”
1–2%	52.11% (37) *
3–5%	40.84% (29)
6–10%	7.05% (5)

* Data are given as a percentage and in parenthesis, the number of replies.

**Table 5 animals-13-00672-t005:** Opinions of pig farmers on the importance of risk factors for tail biting given in the questionnaire. Scale: 1 (Note important at all) to 5 (Very important).

Risk Factors	N	Mean	(±stdv)
Feed quality (over/under-supply of minerals)	77	3.75	1.387
Stocking density	78	3.58	1.559
Ventilation	79	3.44	1.500
Tail length	78	3.35	1.650
High temperatures	78	3.04	1.481

**Table 6 animals-13-00672-t006:** Attitudes of Greek pig farmers on tail docking and pig welfare.

Attitudes *	Disagree	Neutral	Agree
There are other ways than tail docking to prevent tail-biting outbreaks	68.4%	10.1%	21.5%
Tail docking is essential to avoid tail biting	14.8%	8.6%	76.5%
Economic incentives should be provided to raise pigs with intact tails	35.4%	11.4%	53.2%
Piglets feel pain when their tails are docked	70.0%	2.5%	27.6%
Curly tails are indicators of good welfare	34.2%	10.1%	55.7%
Do you believe it is better to dock all tails than to run the risk of tail biting even if it concerns just one bitten pig?	58.2%	7.6%	34.2%
Greek pig market can pay a premium for meat from pigs with an intact tail	91.0%	3.8%	5.2%
Health problems constitute a welfare problem for pig industry	11.1%	17.3%	71.6%
Lack of personnel constitutes a welfare problem for pig industry	9.7%	16.1%	74.2%

* Scale: 1–2 = Disagree, 3 = Neutral, and 4–5 = Agree.

## Data Availability

The data presented in this study are available upon reasonable request from the corresponding author.

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
