# Peer review of "Greek Pig Farmers’ Perceptions and Experiences of Tail Biting and Tail Docking"

_animals, 2023, doi:10.3390/ani13040672_

Round 1

Reviewer 1 Report

General comment

I think the paper presents some interesting results but the authors do not exploit the full potential of the data they have collected. Consequently, I believe that extensive revisions are needed to implement the quality of the work before it can be considered for publication.

In general, I find the following points (detailed below in the detailed comments) to be the ones that deserve the largest revisions:

-        INTRODUCTION

  • A clear description of the main(s) Greek pig farming system(s), to contextualise better the work for the reader is missing. This type of information could also help in understanding come of the differences found with the other countries.

-        MATERIALS AND METHODS

  • A clear description of how the questionnaire was built and structured is missing.
  • The statistical methods adopted must be revised and/or at least, some additional specifications evaluations are needed

-        RESULTS

  • The presentation of the results must be improved: in logical (and consequential) terms, in terms of clarity and above all, in my opinion, the anticipation of the discussion of the results should be avoided. I also think there is some key information missing that should be presented in the results section.

-        DISCUSSION

  • the discussion is generally well done; however, it sometimes refers to data that is not present in the results section and I think this is not appropriate.
  • there is a lack of a statement emphasising the limitations of the study and also how the study may be relevant for the next steps in this field

-        ENGLISH: must be revised by a native speaker, there are some terms out of context and sentences often appear rather awkwardly flowing

-        TABLES AND FIGURES: the quality must be increased, both of the tables, figures and presented captions. In particular, captioning is not sufficient to describe them, while each table and figure should 'stand alone'.

-        SUPPLEMENTARY MATERIALS: I don’t find them useful in the present form.

  • Questionnaire:  I do not consider it appropriate to present the questionnaire in Greek in an international journal. I suggest, therefore, translating it into English so that it can be read by the whole audience.
  • Supplementary tables: I do not know if I am wrong, but they seem to me to be the same tables that were presented in the text. What would be the value of presenting them also as supplementary material?

Detailed comments

Introduction

L37: add a comma after “farming”

L38: what do you mean by sustainability? Which sustainability are you referring to (environmental, economic, or social)?

LL39-45: I would move these sentences after the explanation of WHY tail biting represents a welfare risk for pigs in modern intensive farming (after L47).

L42: I would suggest removing “as a practical management of the problem”

L43: I would suggest replacing “this” with “this practice”

L44: I would suggest replacing “this” with “it”

L47: I would suggest expanding this concept to focus the reader on WHY tail biting is a crucial issue to address for the welfare of the animals

LL41 and 48: are you sure “outbreak” is the correct term to use in this case? It sounds a bit strange to me in this context.

L59: I would suggest replacing “on” with “due to”

Table 1: The authors are not expressed in line with the “Animals” journal requirements for the references. I would also add a column with a brief description of the farming systems in each country. This table, although shows interesting information, might be also moved to supplementary materials to leave more space for tables/figures of the results.

LL73-75: adding a description of the Greek pig farming system(s) (and adding the information in Table 1 to the farming systems of the other countries) would help clarify this sentence. At the moment seems a bit vague.

Materials and methods

LL92-95: I think that there was an error and two sentences might have accidentally been merged

L107-119: I think that considering the available data, another type of statistical approach would have been more suitable (such as multivariate statistical analysis). Thus, I suggest to the authors either change the statistical approach or reflect a bit more on the collected data to try to get as much as they can from it. At the moment I think that the result obtained does not exploit the potential of the information collected. For example, I suggest o

L117: I don’t think that the sentence “search for possible correlations” is adequate for a manuscript. It shows more of a data mining approach rather than targeted analysis to answer research questions.

L119: did you mean p=0.05?

**missing reference to the supplementary material with the questionnaire (that, If kept, should be translated into English)

**missing a clear description of how the questionnaire was developed: who chose the questions? How were the questions to include selected? were questions included to assess the psychological profile of the farmer? what method was used to verify the quality of the data collected?

Results

L122: I would suggest starting the Results section (after the first sentence) with the data contained in Table 2 (and to move LL128-134 here).

L123: in my opinion, the comparison with other studies should be left for the discussion section

L124: how do you know that the “new generation” of farmers answered the online questionnaire?

L125-127: in my opinion, this would be more appropriate for the materials and methods section.

L129: how can you say “well represented”? Please, add in Table 2 the distribution of respondents for each region and each “farm size” category.

L131: shouldn’t it be “Table 2”?

Table 2: I suggest revising it to add the information suggested in the comment above. I think that the caption needs some revision and better specifications as well. Are the data presented the ones from the National Pig Association? You should specify it in the footnote.

L137-141: from what I understand it seems that you have collected information on the experiences that workers had with animals. Have you also collected information on whether the farmers/workers received adequate training on animal welfare? This is a key component of many questionnaires and animal welfare protocols (e.g., Welfare Quality), and I think that analysing the data also considering this variable would add an interesting oversight on this. E.g., were there differences in the % of tail biting, in the perception, in the intervention measured adopted between trained and not trained farmers? And between experienced and not experienced? Multivariate statistical tests would help you in this.

L141: what do you mean by “scarce resource”?

L 142: why is it relevant to specify the breeds? Please, add some information on whether there was a variation in the tail-biting % concerning the farmed breeds.

LL145-146, LL147-248, L169: I think this is for the discussion.

L157: I don’t see this data reported in tables/graphs, what were the other options to “blood on the tail”

Tables 4 and 5: move them to supplementary materials. I would suggest keeping in the manuscript a summarised version of Table 5 – keeping only the first 5 risk factors. The captions should be better specified.

Table 6: This table is not clear to me (what does the “percentage of TB” stands for?) I would suggest representing the information as a histogram.

L180: add n, r and replace P with “p”

L191: replace “in correlation with” with “in relation to”

Figure 1: The quality of the figure is not adequate for publication, in my opinion. Change the label titles to make them legible and keep the details in the footnote (type of test, explanations of the abbreviations), I think that a post-hoc test is missing to understand the difference between farm sizes. In the boxplot, you should also give the information of which data is statistically different from the others (with bars and *). Rephrase and detail better the caption.

LL197-198: please, discuss this data in the discussion

Discussion

In general, English must be revised and there is not a clear discussion of each of the results presented (e.g., LL137-162 are not discussed; L197). The authors do not discuss some of the data presented but discuss others that are not presented in the results section and this creates a lot of confusion.

L209: replace “bitter” with “biter”

L220: replace “of” with “with”

L237: what where the finding of the study held in Finland? Please, discuss these differences between countries considering the farming systems difference.

L258: I think that the term “agenda” is not appropriate

L261-263: expand this section: How could cytokines increase the tail-biting behaviour? Why tail biters can have higher problems such as lung lesions, abscesses and osteomyelitis?

L271: please, specify “greek farmers”

L280: maybe a reflection should be also done on the need for better training on animal welfare for the people who work with animals. Adding the results on this, to see the changes in relation to experience and welfare training, could help in the discussions.

L302-304: Which results are you referring to? I cannot find them in the results section.

L324: “outbreak”?

L320: Add a comment on the possible bias linked with the farmer's perception (education level/training/experience…etc.). Were there any differences between male and female farmers?

LL321-329: you should justify better this part with literature references and further discussion. As it is, it appears rather vague.

L343: add a section with the limitations of the studies and say how this work could contribute to the development of the sector. I would also add some suggestions on what can be done to improve the sector, also taking into consideration farmers’ opinions and difficulties.

Conclusions

L347: you should discuss this data first in the discussion, trying to explain.

L349-352: long and not fluid sentence

L374: title missing

Author Response

General comment

I think the paper presents some interesting results but the authors do not exploit the full potential of the data they have collected. Consequently, I believe that extensive revisions are needed to implement the quality of the work before it can be considered for publication.

In general, I find the following points (detailed below in the detailed comments) to be the ones that deserve the largest revisions:

-        INTRODUCTION

  • A clear description of the main(s) Greek pig farming system(s), to contextualise better the work for the reader is missing. This type of information could also help in understanding come of the differences found with the other countries.

-        MATERIALS AND METHODS

  • A clear description of how the questionnaire was built and structured is missing.
  • The statistical methods adopted must be revised and/or at least, some additional specifications evaluations are needed

-        RESULTS

  • The presentation of the results must be improved: in logical (and consequential) terms, in terms of clarity and above all, in my opinion, the anticipation of the discussion of the results should be avoided. I also think there is some key information missing that should be presented in the results section.

-        DISCUSSION

  • the discussion is generally well done; however, it sometimes refers to data that is not present in the results section and I think this is not appropriate.
  • there is a lack of a statement emphasising the limitations of the study and also how the study may be relevant for the next steps in this field

-        ENGLISH: must be revised by a native speaker, there are some terms out of context and sentences often appear rather awkwardly flowing

-        TABLES AND FIGURES: the quality must be increased, both of the tables, figures and presented captions. In particular, captioning is not sufficient to describe them, while each table and figure should 'stand alone'.

-        SUPPLEMENTARY MATERIALS: I don’t find them useful in the present form.

  • Questionnaire:  I do not consider it appropriate to present the questionnaire in Greek in an international journal. I suggest, therefore, translating it into English so that it can be read by the whole audience.
  • Supplementary tables: I do not know if I am wrong, but they seem to me to be the same tables that were presented in the text. What would be the value of presenting them also as supplementary material?

Authors’ Response: We would like to thank you for your time and effort in reviewing our article. We truly appreciate the thoughtful and constructive feedback you provided.

Detailed comments

Introduction

L37: add a comma after “farming”

Response L37: The requested change was made, thank you.

L38: what do you mean by sustainability? Which sustainability are you referring to (environmental, economic, or social)?

Response L38: Thank you for pointing this out. We added the word “economic” for clarification.

LL39-45: I would move these sentences after the explanation of WHY tail biting represents a welfare risk for pigs in modern intensive farming (after L47).

Response LL39-45: The requested change was made, thank you.

L42: I would suggest removing “as a practical management of the problem”

Response L 42: The sentence “as a practical management of the problem” was deleted.

L43: I would suggest replacing “this” with “this practice”

Response L43: The word “this” was replaced by the word “this practice”.

L44: I would suggest replacing “this” with “it”

Response L44: The word “this” was replaced by the word “it”.

L47: I would suggest expanding this concept to focus the reader on WHY tail biting is a crucial issue to address for the welfare of the animals

Response L47: Thank you for your comment. We expanded this concept by adding a relevant explanatory text.

LL41 and 48: are you sure “outbreak” is the correct term to use in this case? It sounds a bit strange to me in this context.

Response LL41 and 48: The term 'outbreak' is used about tail biting in pigs because it refers to a sudden increase in the number of cases of the behaviour, which can quickly escalate and spread throughout a group if not dealt with promptly.

L59: I would suggest replacing “on” with “due to”

Response L59: The word “on” was replaced by the word “due to”.

 Table 1: The authors are not expressed in line with the “Animals” journal requirements for the references. I would also add a column with a brief description of the farming systems in each country. This table, although shows interesting information, might be also moved to supplementary materials to leave more space for tables/figures of the results.

Response Table 1: Your comment is a fair one. We agree that adding a column with a brief description of the farming system in each country is helpful, and respecting your suggestion, we have now moved it to supplementary materials.

LL73-75: adding a description of the Greek pig farming system(s) (and adding the information in Table 1 to the farming systems of the other countries) would help clarify this sentence. At the moment seems a bit vague.

Response LL73-75: Regarding the additional information on the Greek pig farming system, we added a relevant explanatory text. Regarding addition of Greek pig farming systems in Table 1 there is not relevant published data yet.

Materials and methods

LL92-95: I think that there was an error and two sentences might have accidentally been merged

Response LL92-95: We rephrased the sentence to be more apparent to our readers. Thank you for your comment.

L107-119: I think that considering the available data, another type of statistical approach would have been more suitable (such as multivariate statistical analysis). Thus, I suggest to the authors either change the statistical approach or reflect a bit more on the collected data to try to get as much as they can from it. At the moment I think that the result obtained does not exploit the potential of the information collected. For example, I suggest o

Response L107-119: Authors agree, and we tried to reflect a bit more on the collected data. Thank you for your comment.

L117: I don’t think that the sentence “search for possible correlations” is adequate for a manuscript. It shows more of a data mining approach rather than targeted analysis to answer research questions.

Response L117: Your comment is a fair one; we have now rephrased this.”

L119: did you mean p=0.05?

Response L119: Yes, thank you. An appropriate correction was made.

**missing reference to the supplementary material with the questionnaire (that, If kept, should be translated into English)

Response: Thank you for your comment. The questionnaire has been translated to English

**missing a clear description of how the questionnaire was developed: who chose the questions? How were the questions to include selected? were questions included to assess the psychological profile of the farmer? what method was used to verify the quality of the data collected?

Response L**: Thank you for your comment. We added a relevant explanatory text.

Results

L122: I would suggest starting the Results section (after the first sentence) with the data contained in Table 2 (and to move LL128-134 here).

Response L122: The requested change was made, thank you.

L123: in my opinion, the comparison with other studies should be left for the discussion section

Response L123: Authors agree, and the requested change was made, thank you.

L124: how do you know that the “new generation” of farmers answered the online questionnaire?

Response L124: Your comment is a fair one. We deleted the words “new generation of”.

 L125-127: in my opinion, this would be more appropriate for the materials and methods section.

Response L125-127: Authors agree, and the requested change was made, thank you.

 L129: how can you say “well represented”? Please, add in Table 2 the distribution of respondents for each region and each “farm size” category.

Response L129: Thank you for your comment. In parenthesis, we added the number of replies coming from the main pig-producing regions of Greece.

L131: shouldn’t it be “Table 2”?

Response L131: Following your recommendation, some of the tables have moved to the Supplementary materials, so all tables have new numbering.

Table 2: I suggest revising it to add the information suggested in the comment above. I think that the caption needs some revision and better specifications as well. Are the data presented the ones from the National Pig Association? You should specify it in the footnote.

Response Table 2: Appropriate changes have been made. Thank you for your comment.

L137-141: from what I understand it seems that you have collected information on the experiences that workers had with animals. Have you also collected information on whether the farmers/workers received adequate training on animal welfare? This is a key component of many questionnaires and animal welfare protocols (e.g., Welfare Quality), and I think that analysing the data also considering this variable would add an interesting oversight on this. E.g., were there differences in the % of tail biting, in the perception, in the intervention measured adopted between trained and not trained farmers? And between experienced and not experienced? Multivariate statistical tests would help you in this.

Response L137-141: Regarding your suggestion about training on animal welfare by farmers/workers, we agree that it is a valuable point and could have improved our study. However, during our initial research, we did not come across this specific aspect and thus did not include it in our analysis. We understand that this could have been a limitation of our study.

L141: what do you mean by “scarce resource”?

Response L141: human resources (such as skilled labor or expertise) are of limited availability in the pig industry in Greece.

L 142: why is it relevant to specify the breeds? Please, add some information on whether there was a variation in the tail-biting % concerning the farmed breeds.

Response L142: Thank you for your feedback. We have made the decision to delete this line from our paper as we believe it is not crucial for the overall understanding of our study.

LL145-146, LL147-248, L169: I think this is for the discussion.

Response LL145-146, LL147-248, L169: Thank you for your comment. The requested change was made, thank you.

L157: I don’t see this data reported in tables/graphs, what were the other options to “blood on the tail”

Response L157: Thank you for your comment. A new table has been added to supplementary materials with the relevant information.

 Tables 4 and 5: move them to supplementary materials. I would suggest keeping in the manuscript a summarised version of Table 5 – keeping only the first 5 risk factors. The captions should be better specified.

Table 6: This table is not clear to me (what does the “percentage of TB” stands for?) I would suggest representing the information as a histogram.

Response Table 6: Appropriate changes have been made. Thank you for your comment.

L180: add n, r and replace P with “p”

Response L180: Thank you for your comment. There is no correlation, so we delete the parenthesis.

 L191: replace “in correlation with” with “in relation to”

Response L191: The requested replacement was made, thank you.

 Figure 1: The quality of the figure is not adequate for publication, in my opinion. Change the label titles to make them legible and keep the details in the footnote (type of test, explanations of the abbreviations), I think that a post-hoc test is missing to understand the difference between farm sizes. In the boxplot, you should also give the information of which data is statistically different from the others (with bars and *). Rephrase and detail better the

caption.

Response Figure 1: Your comment is a fair one. We tried to improve the quality of Fig. 1 and corrections were made according to your recommendations. Thank you.

LL197-198: please, discuss this data in the discussion

Response LL197-198: Thank you for your comment. The requested change was made, thank you.

Discussion

In general, English must be revised and there is not a clear discussion of each of the results presented (e.g., LL137-162 are not discussed; L197). The authors do not discuss some of the data presented but discuss others that are not presented in the results section and this creates a lot of confusion.

L209: replace “bitter” with “biter”

Response L209: The word “bitter” was replaced by the word “biter”.

L220: replace “of” with “with”

Response L220: The word “of” was replaced by the word “with”.

L237: what where the finding of the study held in Finland? Please, discuss these differences between countries considering the farming systems difference.

Response L237: Thank you for your comment. We now discussed these differences between countries considering the farming systems difference.

L258: I think that the term “agenda” is not appropriate.

Response L258: Thank you for your comment. The term “agenda” was changed to” have always been a priority for”.

L261-263: expand this section: How could cytokines increase the tail-biting behaviour? Why tail biters can have higher problems such as lung lesions, abscesses and osteomyelitis?

Response L261-263: Authors agree, and we expanded the section on cytokines. We also clarified that tail-bitten pigs have higher problems such as lung lesions, abscesses and osteomyelitis by adding the word “lesions” after “tail biting can cause.”

L271: please, specify “greek farmers”

Response L271: The requested change was made.

L280: maybe a reflection should be also done on the need for better training on animal welfare for the people who work with animals. Adding the results on this, to see the changes in relation to experience and welfare training, could help in the discussions.

Response L280: Thank you for your comment. Although it would help in the discussions we didn’t collect any such data. It will be mentioned as limitation of the study in the relevant section.

L302-304: Which results are you referring to? I cannot find them in the results section.

Response L302-304: Thank you for your comment, as it brought to our attention that the wrong number was mentioned. The results have been added to table 7.

L324: “outbreak”?

Response L324: The term 'outbreak' is used about tail biting in pigs because it refers to a sudden increase in the number of cases of the behaviour, which can quickly escalate and spread throughout a group if not dealt with promptly.

L320: Add a comment on the possible bias linked with the farmer's perception (education level/training/experience…etc.). Were there any differences between male and female farmers?

Response L320: Authors agree that possible bias linked with the farmer’s perceptions probably exists. Still, during our initial research, we did not come across this particular aspect and therefore did not include it in our analysis. We understand that this may have been a limitation of our study.

LL321-329: you should justify better this part with literature references and further discussion. As it is, it appears rather vague.

 Response LL321-329: Reference was added. Thank you for your comment.

L343: add a section with the limitations of the studies and say how this work could contribute to the development of the sector. I would also add some suggestions on what can be done to improve the sector, also taking into consideration farmers’ opinions and difficulties.

Response L343: A new paragraph was added at the end of the discussion with the limitations of the study and some suggestions on what can be done to improve the sector.

Conclusions

L347: you should discuss this data first in the discussion, trying to explain.

Response L347: Authors agree, and results are presented in the discussion. Thank you.

L349-352: long and not fluid sentence

 Response L349-352: Thank you for your comment. We rephrased it accordingly.

L374: title missing

Response L374: The word “References” has been added.

Reviewer 2 Report

1.Questionnaire in Appendix Awritten in Greece should be translatedin English.Readers do not get it.

2. L131:Table 1 is not correct but table 2.

L153,L170,L190, and L204:“Table 3”,“Table 6”,“Table 4”and “Table 7”should be put in the text, respectively.

3. L180: “P<0.005”meaningstatistically significantis not consistent with“no correlation”in L179.

4. L160: What areenrichment materials(3.4%)except chains (67.1%) and plastic objects (29.5%)?

5. Table 5:It should be explainedin the text.

6. Figure 1: The “1,2,3,4,5”of pen hygiene and CD inthe abscissa axisshould beexplained.

7.No explanation ofthe “Mean”in Table 4 and 5 is the most important inadequacyin this paper. Only ranksof “Intervention measure”and “Risk factors”are explained.

Greater than“3”is interpreted as “useful”and “agree”and under than “3”is interpreted as “not useful”and “not agree”in Table 4 and 5, respectively.

This paper may be rejected for publishing if the mean levelsarenot interpreted.

8.Discrepancyshould be discussed betweenthe first 2 ranking(greater than 3: agree) of “Feed quality”and “Stocking density”in the table 5 and lower ranking (lesser than 3: not useful) of “Reduce stocking density”and “Pig lick blocks”in the table 4. 

Author Response

Authors’ Response: We would like to thank you for your time and effort in reviewing our article. We truly appreciate the thoughtful and constructive feedback you provided.

1.Questionnaire in Appendix A written in Greece should be translated in English. Readers do not get it.

Response 1: Thank you for your comment. The Questionnaire has been translated to English.

  1. L131: Table 1 is not correct but table 2.

Response 2. L131: Following the recommendation from reviewer 1, some of the tables have moved to the Supplementary materials, so all tables have new numbering.

L153, L170, L190, and L204:“Table 3”,“Table 6”,“Table 4”and “Table 7”should be put in the text, respectively.

Response L153, L170, L190 and L204: Following the recommendation from reviewer 1, some of the tables have moved to the Supplementary materials, so all tables have new numbering.

  1. L180: “P<0.005” meaning statistically significant is not consistent with “no correlation” in L179.

Response 3: Your comment is a fair one. We have deleted “(spearman rank, p<0.005)”

  1. L160: What are enrichment materials (3.4%) except chains (67.1%) and plastic objects (29.5%)?

Response 4: Thank you for your comment. The other enrichment materials are wood blocks or straw bedding.

  1. Table 5: It should be explained in the text.

Response 5: Appropriate changes made according to your recommendations. Thank you.

  1. Figure 1: The “1,2,3,4,5” of pen hygiene and CD in the abscissa axis should be explained.

Response 6. Figure 1: Your comment is a fair one. We tried to improve the quality of Fig. 1 and corrections were made according to your recommendations. Thank you.

7.No explanation of the “Mean” in Table 4 and 5 is the most important inadequacy in this paper. Only ranks of “Intervention measure” and “Risk factors” are explained.

Greater than “3” is interpreted as “useful” and “agree” and under than “3” is interpreted as “not useful” and “not agree” in Table 4 and 5, respectively.

This paper may be rejected for publishing if the mean levels are not interpreted.

Response 7: Your comment is a fair one. We added the relevant text.

8.Discrepancy should be discussed between the first 2 ranking (greater than 3: agree) of “Feed quality” and “Stocking density” in the table 5 and lower ranking (lesser than 3: not useful) of “Reduce stocking density” and “Pig lick blocks” in the table 4. 

Response 8: Thank you for your comment. A paragraph has added to the discussion for this discrepancy.

Round 2

Reviewer 1 Report

I believe the work has improved considerably, however, I do not understand some of the changes made by the authors, and I do not find in the text some of the changes that, according to the authors' answers, should have been implemented. In my opinion, the work still needs extensive revision work (tables are not mentioned in the text, some are confusing, concepts are not always expressed clearly and are not always supported by adequate bibliographical references). Furthermore, I think that one of the critical points of the work is the statistical approach and consequently the presentation of the results, which is confusing and unclear to the reader.

This general comment is followed by the following specific comments:

LL42-47: references missing for all the added sentences.

L44:”Furthermore, tail biting indicates reduced welfare also for the pigs performing the behaviour” - please, explain how.

LL69-71: A reference is missing to Supplementary Table 1

Supplementary Table 1: I think the format of the table must be improved. The author's citations must be in agreement with the journal’s guidelines for authors. In my opinion the column “main farming system” should be divided in two, keeping in one only the information referred to “indoor/outdoor”, and the in the other columns all the other (bedding use…etc.)

LL80-81: the term “small-medium size” can be relative and the interpretation varies according to the origin of the reader. Please, provide a quantification of the average number of sows/pigs in the typical Greek farm.

LL80-83: reference missing

LL110-112: please, reformulate the sentence

L130: what do you mean by “which does not discriminate differences well in data based on a categorical scale”

LL125-137: The authors responded to my previous comment on statistical analysis by agreeing that other types of statistical tests more suited to the data they have available (e.g. binomial generalised linear models) should be considered in greater depth. However, no changes were observed in the statistical approach which, in my humble opinion, is flawed given the types of data they have to deal with. 

Table 1: according to the journal’s guidelines the table should be below the paragraph where they are cited for the first time. 

Note that the sum of the percentages does not equal 100. 

I do not understand what the data in this table in the brackets refer to, and the explanation does not help to understand it.

Table 3 (and also for the others): the table should be referenced also in the text. In this specific case, it should also be re-formatted 

Table 4: I think this should go logically before table 3 as this consideration appears first in the text.

Table 5: as already suggested, spell-out TB in the footnote. The tables should stand alone.

Figure 1: I can’t see the improvement commented on by the authors in the figure. I still think that this version is not ready for publication (please, see my previous comment). 

L 238: please, move the “.” After the reference

L302: what do you mean by “sickness behaviour”? I get the link between poor health/discomfort with tail biting, but I still have some difficulties in understand the physiological role of cytokines in this. Please, explain.

Reviewer 2 Report

The second version is correctly revised.

Round 3

Reviewer 1 Report

 accept in the present form